# The Role of Prostaglandins in Different Types of Cancer

**DOI:** 10.3390/cells10061487

**Published:** 2021-06-13

**Authors:** Álvaro Jara-Gutiérrez, Victoriano Baladrón

**Affiliations:** Área de Bioquímica y Biología Molecular, Departamento de Química Inorgánica, Orgánica y Bioquímica, Facultad de Medicina de Albacete/CRIB/Unidad de Biomedicina, Universidad de Castilla-La Mancha/CSIC, C/Almansa 14, 02008 Albacete, Spain; alvaro.jara@alu.uclm.es

**Keywords:** prostaglandin, COX-1, COX-2, NSAIDs, cancer

## Abstract

The prostaglandins constitute a family of lipids of 20 carbon atoms that derive from polyunsaturated fatty acids such as arachidonic acid. Traditionally, prostaglandins have been linked to inflammation, female reproductive cycle, vasodilation, or bronchodilator/bronchoconstriction. Recent studies have highlighted the involvement of these lipids in cancer. In this review, existing information on the prostaglandins associated with different types of cancer and the advances related to the potential use of them in neoplasm therapies have been analyzed. We can conclude that the effect of prostaglandins depends on multiple factors, such as the target tissue, their plasma concentration, and the prostaglandin subtype, among others. Prostaglandin D2 (PGD_2_) seems to hinder tumor progression, while prostaglandin E2 (PGE_2_) and prostaglandin F2 alpha (PGF_2α_) seem to provide greater tumor progression and aggressiveness. However, more studies are needed to determine the role of prostaglandin I2 (PGI_2_) and prostaglandin J2 (PGJ_2_) in cancer due to the conflicting data obtained. On the other hand, the use of different NSAIDs (non-steroidal anti-inflammatory drugs), especially those selective of COX-2 (cyclooxygenase 2), could have a crucial role in the fight against different neoplasms, either as prophylaxis or as an adjuvant treatment. In addition, multiple targets, related to the action of prostaglandins on the intracellular signaling pathways that are involved in cancer, have been discovered. Thus, in depth research about the prostaglandins involved in different cancer and the different targets modulated by them, as well as their role in the tumor microenvironment and the immune response, is necessary to obtain better therapeutic tools to fight cancer.

## 1. Introduction

### 1.1. A Piece of History

In 1930, R. Kurzrok and C. C. Lieb demonstrated that the uterine endometrium contracted and relaxed rhythmically after exposure to semen [1]. In 1939, Ulf von Euler stated that this contraction was due to the action of an unknown unsaturated lipid, which he called prostaglandin [2]. Then, S. Bergstrom observed the effects of the administration of prostaglandin E in humans [3]. Between 1962 and 1966, the team of S. Bergstrom and D. A. van Dorp reported having achieved the synthesis of PGE_2_ from arachidonic acid and to have discovered the crystalline structure of PGF_2α_ and PGE_2_, which allowed the synthesis of prostaglandins in sufficient quantities to carry out pharmacological studies [4]. In 1971, J. R. Vane demonstrated that ASA (acetyl salicylic acid, aspirin) and non-steroidal anti-inflammatory agents inhibited the synthesis of prostaglandins [5]. For their research on prostaglandins, S. Bergstrom, B. Samuelsson, and R. Vane, received the Nobel Prize in Medicine and Physiology in 1982.

### 1.2. Structure of Prostaglandins

Prostaglandins (PGs) are a family of lipids of 20 carbon atoms derived from polyunsaturated fatty acids, especially arachidonic acid (AA). Carbon atoms C8-C12 form a cyclopentane ring, while carbons C1-C7 and C12-C20 constitute two parallel aliphatic chains (R1 and R2, respectively) [6,7]. PGs are not produced by specialized glands but by different types of cells in the body, acting as autocrine and paracrine messengers. In addition, their half-life is short. Along with thromboxanes, they are known as prostanoid lipids, and share the same carbon skeleton. In turn, they are part of a larger group called eicosanoids, along with leukotrienes, lipoxins and hydroeicosatheroid and epoxyeicosatrieneic acids [8].

Regarding the nomenclature of PGs, it is summarized as follows: the accompanying letter to the acronym PG indicates the components of the cyclopentane ring and its solubility in different solvents (A: unsaturated ketones, soluble in cold acetone; E: β-hydroxyketones, soluble in ether; F: 1.3-dioles, soluble in phosphate buffer); the numeric subscript indicates the number of double links present in parallel strings; and the symbolic subscript indicates other structural details (α: OH group of C9 located on the same plane as the ring of R1; β: OH group in a plane other than R1) [6,7,9].

### 1.3. Prostaglandin Synthesis

PGs derive from AA oxidation, a process catalyzed by enzymes called cyclooxygenases (COX) or PGH (prostaglandin H) synthetases. In humans, there are two isoforms, COX-1 and COX-2 [10,11,12]. Both share 60% of their amino acid sequence and have a similar three-dimensional structure, although COX-1 and COX-2 are encoded by different chromosomes (chromosomes 9 and 1, respectively) (2). COX enzymes are membrane proteins with a molecular weight of approximately 70–74 kDa and possess four protein domains: (1) amino-terminal signal peptide domain, which guides the protein to its destination in the membranes. This domain has a longer length and greater hydrophobicity in the COX-1 isoform; (2) dimerization domain, which allows the stability of the protein by means of disulfide bridges, which link it to the catalytic domain, and salt bridges; (3) membrane binding domain, formed by four amphipathic helices that allow its insertion into the lipid bilayer; (4) catalytic domain (carboxyl-terminal), which has two separate catalytic sites with different functions (peroxidase and cyclooxygenase activities). Within this catalytic domain, there is an arginine residue at position 120 (Arg120) essential for substrate binding and inhibitory NSAIDs (non-steroidal anti-inflammatory drugs) action in COX-1, but it does not seem to be as determinant in COX-2 [13,14,15,16].

The COX enzymes mature in the endoplasmic reticulum, and they are then transported and inserted into the cell membrane, forming functional homodimers, with an access channel that allows the entry of AA. Precisely, ASA acetylates a serine residue on the surface of the access channel, irreversibly preventing the passage of the substrate, thus stopping the chain of chemical reactions [17,18]. The COX-1 enzyme is constitutively expressed in most tissues of the human body and is responsible for regulating the basal homeostasis of prostanoid synthesis, whereas the COX-2 enzyme promotes the transient and inducible synthesis of these compounds when eventual physiological or pathological processes occur, responding to certain stimuli. Among these stimuli are interleukin 1 (IL1), fibroblast growth factor (FGF), tumor necrosis factor (TNF), bacterial lipopolysaccharide (LPS) and nuclear transcription factor NFκB [10]. However, basal expression of COX-2 has been demonstrated in the brain, testis, tracheal epithelium and in the macula densa region of the kidneys [10].

AA is a polyunsaturated acid consisting of 20 carbon atoms, found in the phospholipids of the cell membrane. By the action of different extracellular stimuli (bradykinin, adrenaline, thrombin, corticoids, among others), phospholipases A2 and C are activated in the first phase of prostaglandin synthesis [19]. Phospholipase A2 directly releases AA from the phospholipids phosphatidyl-choline and phosphatidyl-ethanolamine from the cell membrane. Phospholipase C releases diacylglycerol (DAG) from phosphatidyl inositol, and from the DAG generated, AA is released through the action of the enzyme diglyceride lipase. AA then travels through the cytosol until it is metabolized by the cyclooxygenase pathway [20], which generates prostaglandins and thromboxanes (TXA), by the lipoxygenase pathway, generating leukotrienes (LTB) and lipoxins (LXA) [7], or by the cytochrome P450 pathway, generating hydroeicosatrienoic acids (HETE) and epoxyeicosatrienoic acids (EET) [21].

Since the focus of this review is on PGs and the COX pathway, the remaining two pathways of AA-derived products are briefly described below [22]. The lipoxygenase (LOX) pathway gives rise to LTB and LXA. The enzymes required for LTB synthesis are present in leukocytes, macrophages, and mast cells, while those required to form LXA are found in leukocytes and platelets. Hyperactivity of this pathway has been associated with rheumatoid arthritis, inflammatory bowel disease, allergic rhinitis, bronchial asthma, osteoarthritis, and atherosclerosis. On the other hand, the cytochrome P450 pathway allows obtaining the EETs thanks to the CYP2C (P450 arachidonic acid epoxygenase 2C) and CYP2J (P450 arachidonic acid epoxygenase 2J) enzymes, and the HETEs by means of the CYP4A (cytochrome P450 4A fatty acid omega hydroxylase) enzyme activity. From EETs, dihydroxyeicosatrienoic acids (DHETEs) are generated by the action of the soluble enzyme epoxide hydrolase (sEH). All these acidic compounds allow the maintenance of vascular homeostasis by acting as vasodilators or vasoconstrictors depending on which compound they are transformed into [8].

In the cyclooxygenase pathway, a series of reactions catalyzed by COX1/2 enzymes culminate in the synthesis of the different types of PG and TXA. First, peroxidation leads to prostaglandin G2 (PGG_2_), an unstable and short-lived product. Then, deperoxidation generates prostaglandin H2 (PGH_2_), a direct precursor of other PGs (PGE_2_, PGD_2_, PGF_2_ and PGI_2_) and thromboxanes such as TXA_2_, which are synthesized by different enzymes depending on the tissue and the physiological state [7].

### 1.4. Mechanism of Action of Prostaglandins. Transport and Degradation

PGs cross the cell plasma membrane of cells into the extracellular medium by facilitated diffusion. PGs are anions at physiological pH, and as the intracellular voltage is about 20 times more negative inside the cell than outside, the tendency is to exit to the extracellular space. In addition, there is a concentration gradient of PGs; inside the cell their concentration is higher than outside, thus favoring their outflow. The magnitude of the effects of PGs depends not only on their synthesis, but also on their degradation, preferentially in the lung. The elapsed time until they are catabolized is usually seconds, preventing them from reaching the general circulation. Once they have fulfilled their function, PGs are carried into the cell by the prostaglandin transporter (PGT, SLCO2A1) [23]. Once inside, they are oxidized and inactivated by the enzyme 15-hydroxy-prostaglandin dehydrogenase (15-PGDH). The final product is eliminated in the urine [24].

Most of the actions of prostanoids are mediated by receptors coupled to different G (guanine nucleotide-binding protein) proteins that possess seven transmembrane domains and are encoded by different genes. Each of the prostanoids has at least one different receptor. Regarding PGs, eight different receptors have been described: two receptors for PGD_2_ (DP1 and DP2), four for PGE_2_ (EP1, EP2, EP3 and EP4), one for PGF_2α_ (FP) and one for PGI_2_ (IP) [25]. The IP, DP1, EP2 and EP4 receptors mediate an increase in cAMP (cyclic adenosine monophosphate) and have been termed “relaxant-type receptors”, whereas the group composed of the FP and EP1 receptors associated with Gq (G protein heterotrimeric that activates beta isoforms of phospholipase C) induce calcium mobilization and are termed “contractor-type receptors”. In addition, FP receptors may be associated with RHO (RAS homologous GTPase protein) protein bound to small Gs (G protein that stimulates the cAMP-dependent pathway by activating adenylyl cyclase) protein via a Gq-independent pathway [26]. The DP2 receptor belongs to a different subgroup and is considered a member of the “chemoattractant receptors” subgroup. DP2 is associated with inhibitory Gi (G protein that transmits an inhibitory signal from membrane receptors to adenylyl cyclase) protein inhibiting cAMP synthesis and increasing intracellular Ca^2+^ concentration. The EP3 receptor, formerly called “inhibitory receptor”, can bind to Gi or G12 (G protein that link cell surface G protein-coupled receptors primarily to guanine nucleotide exchange factors for the RHO small GTPases) and causes a decrease in intracellular cAMP levels, increases Ca^2+^ concentration and activates the RHO protein associated with a family of small G proteins [26].

### 1.5. General Functions of Prostaglandins

The prostaglandin functions depend on the organ or tissue, the receptor to which they bind, and the physiological situation. Broadly speaking, these functions are shown in Table 1. It could be concluded that they help to maintain homeostasis of the different organs/tissues and that they compose an alert system to normal and pathological physiological processes, giving rise to the appearance of inflammatory signs and pain, among many others [27].

### 1.6. Prostaglandin Inhibitors

There are different substances capable of preventing the formation of PGs by inhibiting COX1/2 enzymes [28]. These substances interact with specific amino acids of the enzymes to inhibit them. For example, ASA irreversibly binds to a serine-530 residue of the substrate entry channel, preventing the entry of arachidonic acid into the active site of COX1/2 enzymes that result irreversibly inactivated, especially COX1 [29,30]. Other drugs, such as ibuprofen and naloxen, act as competitive inhibitors of arachidonic acid and are more specific of COX2 enzyme [30]. For its part, indomethacin produces a time-dependent inhibition of both isoforms thanks to the electrostatic interaction between its carboxyl group and arginine-120 residue of the COX enzymes channel [31]. The presence of two valine amino acids in COX-2 (isoleucine in COX-1) allows some inhibitors to act selectively on COX2, thus avoiding the gastrointestinal and renal side effects typical of COX-1 inhibition [32]. Among the more selective COX-2 inhibitors, we find etoricoxib, lumaricoxib, nimesulide, among others.

## 2. Cancer and Prostaglandins

Previously, it has been mentioned that PGs are involved in a wide variety of biological processes. Within these processes, it is essential to include research on the involvement of these compounds in the development of different types of neoplasms, which is the objective of this review. We have revised many published works with interesting data about prostaglandins that are involved in different types of cancer, and we have described some of the latest therapeutic advances to treat cancer by acting on prostaglandins and enzymes related to these neoplasms.

### 2.1. Prostaglandins in Skin and Bone Cancer

Skin cancer most often develops on skin exposed to the sun, but it can also occur on areas of the skin not ordinarily exposed to sunlight. There are three major types of skin cancer: basal cell carcinoma, squamous cell carcinoma, and melanoma, which have different neoplastic characteristics, some of them related with prostaglandin actions.

In one study, a high expression of the enzyme aldo-keto reductase 1C3 (AKR1C3^+^) was found in skin squamous cell carcinoma, which reduces PGD_2_ levels by metabolizing it to PGF_2α_. This same study showed that PGD_2_ inhibits the formation of new vessels and, therefore, its reduction facilitates the neovascularization necessary for tumor progression [33].

Another study demonstrated that in this type of cancer, an increase in miR-31-5p micro–RNA is needed, which generates a decrease in acyl-coenzyme A peroxisomal A oxidase 1 (ACOX-1), an enzyme that favors normal concentrations of different lipids, including PGs. Suppression of ACOX-1 is associated with elevated PGE_2_ concentrations and increased tumor cell migration and invasion. In fact, it is proposed to use PGE_2_ concentrations in saliva as a biomarker of disease progression, since the higher the stage, the higher the concentration of PGE_2_ in saliva [34].

The treatment of squamous cancer and other cancer with 15-deoxy-delta-12,14-prostaglandin J2 (15d-PGJ_2_) has been shown to reduce cell growth, secondary to a lower concentration of the oncogenic STAT3 (signal transducer and activator of transcription 3) factor [35]. Similarly, it has been described that at higher concentrations of prostacyclin (PGI_2_), the 5-year survival rate in this type of cancer is higher. The increase of this prostaglandin would influence cell proliferation, cell migration and the inflammatory process [36].

A work carried out with mice showed that treatment of non-melanoma type cancer with apigenin, a compound found in fruits and vegetables, inhibits its progression. This substance produces inhibition of tissue polypeptide antigen (TPA), a tumor inducer driven by exposure to ultraviolet B (UVB) radiation, and a reduction in the concentration of COX-2, PGE_1_, and EP1 and EP2 receptors [37]. Another study has also demonstrated the efficacy of piroxicam as a preventive agent, as it also reduces COX-2 expression. Moreover, the local use of piroxicam on actinic keratoses and field cancerization has also been reported, confirming its efficacy as target therapy [38].

Elevated concentrations of PGF_2α_ have also been found in melanoma tumor cells with respect to healthy tissues [35]. In cell and mouse models, this prostaglandin interferes with the mechanism of action of ASA. ASA blocks the expression of the sex-determining region of the Y chromosome (SRY) related to high mobility group 2 (SOX2), thereby promoting cell apoptosis. Increased PGF_2α_ rescues such cells from cell death. Moreover, the use of antagonists of this prostaglandin potentiates the action of ASA [39].

Melanotan II (MTII), a synthetic analogue of the alpha-melanocyte stimulating hormone (alpha-MSH), potently inhibited the migration, invasion, and colony-forming capability of B16-F10 melanoma cells in vitro and in vivo despite a lack of influence on proliferation [40]. MTII treatment inhibited COX-2 expression and PGE_2_ production via PTEN (fosfatidilinositol-3,4,5-trisfosfato 3-fosfatasa) upregulation, thereby suppressing melanoma progression. Hence, topical MTII therapy may facilitate a novel therapeutic strategy against melanoma.

On the other hand, overexpression of staphylococcal nuclease domain containing 1 (SND1) has been found in several malignancies including osteosarcoma, which is the most frequent primary bone tumor. Zhou and coworkers revealed that osteosarcoma tissues from different patients expressed significantly high SND1 mRNA and protein expression compared to normal bone tissues. They found that SND1 overexpression significantly promoted cell proliferation and tumor growth in vitro cell lines and *in vivo*. Their results also revealed that SND1 increased the production of PGE_2_. The serum PGE_2_ level had a significant positive association with the SND1 mRNA expression level in osteosarcoma tissues. Additionally, they found that SND1 upregulated PGE_2_ expression through the NFκB/cyclooxygenase2 (COX2) pathway. Targeting of SND1 as a new antitumor strategy for patients with osteosarcoma and SND1 may also act as a potential biomarker of the therapeutic strategies utilizing COX2 inhibitors [41].

Recent studies showed that the activation of prostaglandin receptor EP1 promotes cell migration and invasion in different types of cancer, including osteosarcoma. Niu and coworkers investigated the role of EP1 in the proliferation of osteosarcoma cells in vitro and in vivo. EP1 levels were significantly higher in osteosarcoma cells compared to osteoblasts. PGE_2_ or 17-PT-PGE_2_ (17-phenyl-trinor-prostaglandin E2) treatment increased the proliferation and decreased the apoptosis of cells. Inhibition of EP1 by SC51089 or siRNA markedly decreased the viability of cells. EP1 appears to be involved in PGE_2_-induced proliferative activity of cells. Antagonizing EP1 may provide a novel therapeutic approach to the treatment of osteosarcoma [42].

### 2.2. Prostaglandins in Breast Cancer

Breast cancer is a commonly reported cancer that is widely prevalent among women. Its early detection improves patient survival and results in better outcomes. For diagnosis and follow-up care, tumor markers are one of the feasible investigations to be ordered. 8-Iso-prostaglandin F_2a_ (8-iso-PGF_2α_) serves as a serum non-invasive marker reflecting oxidative stress and subsequent damaging of DNA. The serum level of 8-iso-PGF_2α_ in the breast cancer patients (57.92 pg/mL) was significantly higher compared to those with benign tumors (18.89 pg/mL) (*p* < 0.001) [43].

Currently, a preventive treatment for this type of cancer is tamoxifen, which is only effective in cases of estrogen receptor (ER)-positive cells. A published review proposes the use of NSAIDs as chemopreventive agents for this type of cancer due to their multiple interactions with tumor development. NSAIDs inhibit tumor DNA synthesis, modulate ER concentration and the COX, NF-κB, caspases and WNT (wingless and Int-1)-β-catenin-TCF4 (transcription factor 4) pathways, as well as glycogenesis, by inactivating 6-phosphofructose-1-kinase) [44].

The aldo-keto reductase (AKR) superfamily is gaining attention in cancer research. AKRs are involved in important biochemical processes and have crucial roles in carcinogenesis and chemoresistance. The enzyme AKR1C3 has many functions, which include production of prostaglandins, androgens and estrogens, and metabolism of different chemotherapeutics; AKR1C3 is thus implicated in the pathophysiology of different types of cancer, including breast cancer. The actions of AKR1C3 can produce FP receptor ligands whose activation results in carcinoma cell survival. 11β-Prostaglandin F2α, a bioactive metabolite catalyzed by AKR1C3, stimulates prostaglandin F receptor, and induces slug expression in breast cancer [45]. It has been demonstrated that a high concentration of the PGF_2α_-bound FP receptor is significantly (*p* = 0.02) related to a higher level of Ki-67 (nuclear protein related to cell proliferation) in AKR1C3^+^ breast cancer patients. Treatment of cells with an AKR1C3 inhibitor reduces PGF_2α_ levels comparable to those in healthy tissues. With squamous skin cancer, a study found that high concentrations of PGI_2_ corresponded with a reduction in survival time. Moreover, it has been determined that this prostaglandin activates the ERK1/2-MAPK (extracellular signal-regulated kinase 1/2 mitogen-activated protein kinase) and NF-κB pathways by binding to the FP receptor, leading to increased resistance to chemotherapy treatment. This activation is slowed by using inhibitors of FP (AL8810) and NF-κB (BAY 11e7082 and Parthenolide) [45].

As described above, a study described the activation of the AKT (protein kinase B (PKB)-AP1 pathway in breast cancer by treatment with high concentrations of 15d-PGJ_2_. When this pathway is irregularly activated, it is normally modulated by the tumor suppressor PTEN, which acts by dephosphorylating PIP3 (phosphatidylinositol (3,4,5)-triphosphate). This study demonstrated that prostaglandin 15d-PGJ_2_ covalently interacts with the cysteine-136 residue of PTEN and modifies it in such a way that it loses its ability to inhibit the AKT-AP-1 pathway [46]. On the other hand, some authors have shown that dihomo-γ-linoleic acid (DGLA) can act as an inhibitor of the COX-2 enzyme by inhibiting the enzyme delta-5-desaturase, which converts DGLA into arachidonic acid. Once inhibited, DGLA acts as a COX-2 substrate and is degraded to 8-hydroxyoctanonic acid that activates the caspases and poly-ADP-ribose polymerase (PARP) pathways. In addition, inhibition with DGLA has been shown to increase the efficacy of 5-fluorouracil treatment, which decreases tumor migration and invasion [46].

A recent study has shown that the increased concentration of the PGE_2_-EP2 complex limits the expression of the membrane protein CD80 (cluster of differentiation 80) in macrophages, which hinders macrophages polarization and the immune system (IS) response to cancer in human and mouse cells. Knock-out mice of microsomal PG synthetase 1 (mPGES-1) eliminates this limitation [47]. On the other hand, an in vivo experiment with different mouse models showed that ibuprofen reduces PGE_2_ levels and tumor volume in a dose-dependent manner, associated with an increase in mature macrophages, increased recruitment of CD45 (leukocyte common antigen)^+^ T lymphocytes, and decreased numbers of immature monocytes [48].

The results of a case-control study with human breast cancer patients in a phase-II randomized trial suggest that the perioperative use of β-antagonists such as propranolol, and COX-2 inhibitors such as etodolac, can significantly block STAT and EGR3 (early growth response protein 3) pathways, which would favor tumor dissemination. Regarding cellular activity, an increase in NK (natural killer) lymphocytes and B cells associated with a lower number of monocytes. Also, plasma interleukin 6 (IL-6) levels decreased [49].

In inflammation-associated carcinogenesis, COX-2 is markedly overexpressed, resulting in accumulation of various prostaglandins with oncogenic potential such as 15d-PGJ_2_. The epithelial-to-mesenchymal transition (EMT) is a process by which epithelial cells lose their polarity and adhesiveness, and thereby gain migratory and invasive properties. Treatment of human breast cancer MCF-7 cells with 15d-PGJ_2_ induced EMT as evidenced by increased expression of Snail (zinc finger transcriptional repressor) and ZEB1 (zinc finger E-box-binding homeobox 1), with concurrent down-regulation of E-cadherin and production of CXCL8 (chemokine (C-X-C motif) ligand 8) as a putative activator of fibroblasts, which may contribute to tumor-stroma interaction in inflammatory breast cancer microenvironment [50].

On the other hand, it is known that the formation of new blood (angiogenesis) and lymphatic (lymphangiogenesis) vessels are major events associated with most epithelial malignancies, including breast cancer. Inflammation is a key mediator of angiogenesis and lymphangiogenesis with aberrant expression of COX2, which is a major promoter of both events by the activation of prostaglandin E receptor EP4 on tumor cells, and the induction of oncogenic microRNAs. The COX-2/EP4 pathway also promotes additional events in breast cancer progression, such as cancer cell migration, invasion, and the stimulation of stem-like cells. EP4 antagonists hold a major promise in breast cancer therapy in combination with other modalities including immune check-point inhibitors [51].

### 2.3. Prostaglandins in Lung Cancer

Lung cancer, also known as lung carcinoma, is a malignant lung tumor characterized by uncontrolled cell growth in lung tissues. This growth can spread beyond the lung by the process of metastasis into nearby tissue or other parts of the body. Most cancer that starts in the lung, known as primary lung cancer, are carcinomas. The two main types are small-cell lung carcinoma (SCLC) and non-small-cell lung carcinoma (NSCLC).

Long-chain acyl-CoA synthetase (ACSL3) channels AA into phosphatidylinositol to provide the lysophosphatidylinositol-acyltransferase 1 (LPIAT1) with a pool of AA to sustain high prostaglandin synthesis. By using lung cancer cell lines, mouse models, some authors show that the LPIAT1 knockdown suppresses proliferation and anchorage-independent growth of non-small cell lung cancer by using cell lines and hinders in vivo tumorigenesis. In primary patient-derived frozen lung adenocarcinoma samples, the expression of LPIAT1 is elevated compared with healthy tissue and predicts poor patient survival [52].

TNF-α (tumor necrosis factor alpha) has been confirmed to promote tumor growth in laryngeal carcinoma. A work found a crosstalk between PGE_2_ and TNF-α signaling pathways, and the interaction between GRK2 (G-protein-coupled receptor kinase 2) and TRAF2 (TNF receptor associated factor 2), which leads to the activation of TNF-α–TRAF2-MMP (cell matrix metalloprotease)-9 signaling, and results in the progression of laryngeal carcinoma [53].

It has been shown both in lung and esophageal cancer cell lines and lung adenocarcinoma biopsy samples a link between activation of the CRTC1 (CREB-regulated transcriptional coactivator 1) oncogene, inactivation of the tumor suppressor LKB1 (hepatic kinase B1), and the presence of glycosylated COX-2 in pulmonary adenocarcinoma. CRTC1 potentiates the cAMP/CREB (transcription factor that binds to DNA sequences “cAMP response elements”) intracellular signaling pathway, which promotes tumor development. By using COX-2 inhibitors such as niflumic acid (NS-398), tumor growth was reduced in CRTC1^+^/LKB1^−^ cases. The glycosylated COX-2-synthesized prostaglandin PGE_2_ binds to the EP2 y EP4 receptors and causes dephosphorylation of CRTC1, which in turn leads to increased COX-2 activation [54].

Other authors have found that high levels of the microRNA miR-574-5p are related to elevated concentrations of the prostaglandin PGE_2_.both in lung adenocarcinoma cell line and lung non-tumor and tumor tissue samples. This microRNA acts as an inhibitory substrate of the ARN1-binding protein in the CUG triplet repeat (CUGBP1), which under physiological conditions suppresses PEGS-1 (PG synthetase 1). Furthermore, the upregulation of this microRNA is associated with high serum levels of interleukin 1β (IL-1β), typical in cases of non-small cell lung cancer with poor prognosis. The use of PGES-1 inhibitors abrogates the effect of miR-574-5p [55].

On the other hand, it has been shown that treatment of human non-small cell lung cancer cells with prostaglandins 15d-PGJ_2_ and PGD_2_ results in increased apoptosis due to increased ROS (reactive oxygen species) synthesis and activation of the caspases pathway [34]. Also, overexpression of the enzyme prostacyclin synthetase (PGIS) in lung cancer cell lines and mouse models has been shown to inhibit MHC-II (major histocompatibility complex-II)^+^ lung cancer growth, independently of PGI_2_, through recruitment of T-CD4 (cluster of differentiation 4)^+^ lymphocytes [56].

A randomized clinical trial with 5888 patients demonstrated that the use of NSAIDs leads to a reduction in lung tumor mass and fewer metastases [57]. Also, a retrospective case-control analysis to determine the effect of NSAID use on the incidence of lung cancer identified 1038 patients with lung cancer from a review of pathology data at several large hospital centers. The study suggested that smokers who regularly use NSAIDs might benefit from a possible protective effect against lung cancer [58].

### 2.4. Prostaglandins in Liver Cancer

There is a well-known relationship between the appearance of hepatocarcinoma and the proliferation of stellate cells. Stellate cells produce a dysregulation of the immune system, generating a decrease in regulatory T lymphocytes and an increase in myeloid-derived suppressor cells (MDSC), which favors tumor progression.

One study performed with cell lines and mouse models confirmed that the increase in stellate cells is due to increased activation of the COX2-PGE_2_-EP4 pathway in these cells [59]. These authors used COX-2 inhibitors such as SC-236 to inhibit the accumulation of stellate cells and stop the spread of liver cancer. The use of EP4 inhibitors, such as AH23848 was also effective, but only in in vitro tests.

Another study showed that prostaglandin PGE_2_ increases MYC (myelocytomatosis and tumor transcription factor) oncogene concentration by activating the EP4-protein-G-Adenylate cyclase-cAMP-kinase A-CREB pathway. This increase is not sufficient to trigger the onset of cancer, but it does facilitate its spread [60].

2,5-Dimethylcelecoxib (DMC) is a targeted inhibitor of mPGES-1, a key enzyme in the PGE_2_ synthesis pathway of inflammatory mediators and the inhibition of growth of hepatitis B virus (HBV)-related hepatocellular carcinoma (HCC). DMC promotes HBV-related HCC immune microenvironment, which not only enrich the relationship between inflammatory factors (mPGES-1/PGE_2_ pathway) and the immunosuppression programmed death-ligand 1 (PD-L1), but also provide an important strategic reference for multitarget or combined immunotherapy of HBV-related HCC. By using mouse models and human tissue samples, these authors demonstrated that DMC combined with atezolizumab had more significant antitumor effect and stronger blocking effect on PD-L1 pathway [61].

### 2.5. Prostaglandins in Digestive System and Pancreas Cancer

Digestive cancer can develop along the entire digestive tract (esophagus, stomach, small intestine, large intestine, anus) as well as in other organs, such as the liver, pancreas, and biliary tracts. Among them, the number of colorectal cancer patients is increasing worldwide.

In a study based on human patients with esophageal squamous cancer, an elevated level of a subtype of Zinc transporter ZIP5 was observed. Inhibition of ZIP5 reduces metastasis and levels of cyclin D1 and COX-2 [62]. Host immunity plays a vital role in tumorigenesis, including tumor invasion and metastasis. The enzyme 15-PGDH, which plays a key role in prostaglandin degradation, is a critical inflammatory mediator in gastric cancer tumorigenesis. A study with gastric carcinoma patients demonstrated that 15-PGDH may contribute to anti-tumor immunity by regulating FOXP3 (fork head box protein 3) [63]. Also, a study with 277 patients showed that the presence of the prostaglandin PGD_2_ reduces tumor expansion of gastric adenocarcinoma by inhibiting the peroxisome proliferator-activated receptor gamma (PPARγ) pathway [36]. An interesting study proposes a connection between *Helicobacter pylori* infection and the development of gastric adenocarcinoma. Infection triggers an increase in COX-2 concentration, which favors the onset of tumor expansion. Prophylaxis based on the use of NSAIDs has been shown to be effective, considering the risk of gastro- and cardiotoxicity. In addition, it has been confirmed that the use of selective cyclooxygenase type 2 inhibitors (COXIB) can stop this gastric adenocarcinoma growth [62].

COX-2, activated in response to inflammatory stimuli, is also one of the major molecules that is involved in the development and progression of colorectal cancer. It has been shown that COX-2 inhibitors prevent the carcinogenesis and help in the treatment of sporadic or familial cases as shown by an overall increase in the survival rate. However, prolonged use of these inhibitors is associated with an increase in the risk of development of cardiovascular complications [64]. One study performed with cell lines, human tissues and mouse models showed that upregulation of the co-activator CRTC1 stimulates expression of the COX-2 gene and the production of prostaglandin PGE_2_, which in turn leads to dephosphorylation and activation of CRTC1, closing the feedback loop that increases tumorigenicity of colorectal cancer [65]. Another study performed in cell lines proposes that the XRCC5 (X-ray repair cross complementing 5) protein, involved in DNA repair and telomere maintenance, and the transcription factor p300, which phosphorylates the XRCC5 gene, potentiate COX-2 expression. XRCC5/p300/COX-2 pathway could be a new therapeutic target since the use of p300 inhibitors seems to reduce COX-2 expression [66]. 

Several case-control and cohort studies demonstrate that continued prevention with ASA reduces the incidence of adenomas and mortality from this colorectal cancer. The effects of ASA begin to be significant after 3 years from the start of the treatment and with doses between 75 and 325 mg/day, with a greater impact on the proximal region of the colon [67]. A randomized clinical trial with 14,743 patients demonstrated that the use of NSAIDs achieves a reduction in tumor mass and a lower number of metastases in this type of cancer [57]. Similarly, another study with stage III patients treated with ASA resulted in a decrease in mortality and neoplastic recurrence of colorectal cancer [62]. In addition, patients receiving combined treatment with ASA and statins were found to have lower levels of PG compared to those receiving only one of the treatments [68]. Another study with mouse models suggests that the use of sulindac in 15-PGDH knockouts could be a better alternative to other NSAIDs, since it reduces the number of new adenomas in the colon, although it is associated with greater inflammatory lesions in the proximal colon. Also, the knock-out of 15-PGDH has been associated with increased resistance to aspirin and celecoxib [69]. Finally, the treatment of colorectal cancer tumor cells with IP6 (inositol hexaphosphate) decreases COX-2 mRNA expression and activates genes of the lipoxygenase pathway, which reduced the concentration of PGE_2_ [70].

On the other hand, a study in different cell lines demonstrated that treatment with inhibitors of the PGE_2_-EP4 complex reduces the formation of new adenoma-like premalignant lesions in the colon, with reduced activation of the PI3K (phosphatidylinositol 3-kinase)-AKT-mTOR (muscular target of rapamycin) and ERK1/2-MAPK signal transduction pathways [71]. The prostaglandin PGF_2α_ has been shown to increase the migration and invasiveness of colorectal tumor cells [45]. However, the treatment of cells with 15d-PGJ_2_ results in the inhibition of telomerase, through modulation of the MYC oncogene, which directs these cells to a state of senescence or cell death [35]. A genetic study with cell lines and immunodeficient mice showed that tumor progression in colorectal cancer is related to the overexpression of the enzyme prostaglandin synthase E, due to an increase in the EGR1 (early growth response protein 1) factor, produced by the action of prostaglandin PGF_2β_. When a PGF_2β_ receptor antagonist was used, EGR1 and PGE_2_ concentrations were reduced [72]. It has been also proposed that the urinary PG urinary metabolite (PGM) may be an early diagnostic marker of colon cancer, since its levels are higher in patients already diagnosed with this pathology, followed by patients with multiple adenomas [73].

Accumulating evidence has shown that the tumor microenvironment, including macrophages, neutrophils, and fibroblasts, plays an important role in the development and progression of colorectal cancer. Although targeting the TME could be a promising therapeutic approach, the mechanisms by which inflammatory cells promote colorectal cancer tumorigenesis are not well understood. Therapies targeting the specific downstream molecules of PGE_2_ signaling could be a promising approach [74].

As commented above for colon cancer, a study in different cell lines, including human pancreatic carcinoma cell line PANC-1, demonstrated that prostaglandin E2 activates the mTORC1 pathway through an EP4/cAMP/PKA- and EP1/Ca^2+^-mediated mechanism [71]. The pancreatic tumor stroma is composed of phenotypically heterogeneous cancer-associated fibroblasts with both pro- and anti-tumorigenic functions. Calcipotriol decreased cancer-associated fibroblasts proliferation and migration and reduced the release of the pro-tumorigenic factors such as prostaglandin E2 in cancer-associated fibroblasts (CAFs) isolated from human pancreatic tumor tissues. However, calcipotriol promoted PD-L1 upregulation, which could influence T cell mediated tumor immune surveillance and T cell activation [75]. Vitamin D3 analogues appear to have dual functions in the context of pancreatic cancer, which could have important clinical implications.

Several studies have shown that pancreatic cells do not express COX-1, which explains why ASA does not adequately prevent disease progression. Therefore, selective COX-2 inhibitors, such as COXIB-2 (celecoxib), should be used to reduce elevated COX-2 activity. However, COXIB-2 inhibitors have not demonstrated comparable efficacy to cisplatin and gemcitabine as a treatment for this type of cancer [62].

### 2.6. Prostaglandins in Renal and Urinary Cancer 

Almost all kidney cancers are renal cell carcinomas. Most solid kidney tumors are cancerous, but purely fluid-filled tumors (cysts) generally are not. Exposure to cadmium (Cd) is considered to be a threat to human health. The kidney is the main target of Cd accumulation, which increases the risk of renal cell carcinoma. Shi and coworkers suggested that cells exposed to low dose Cd promoted migration of renal cancer cells, which was not dependent on Cd-induced reactive oxygen species (ROS) and intracellular Ca^2+^ levels. Cd exposure induced cAMP/PKA-II (protein kinase A type 2)-COX2, which mediated cell migration and invasion, and decreased expressions of the EMT marker, E-cadherin, but increased expressions of N-cadherin and Vimentin. This study might contribute to understanding of the mechanism of Cd-induce progression of renal cancer and future studies on the prevention and therapy of renal cell carcinomas [76]. It has also been shown that the higher the expression of COX-1, the higher the degree of malignancy of the renal tumor [36].

The use of 15d-PGJ_2_ in patients undergoing radical nephrectomy for renal cell carcinoma caused apoptosis of tumor cells through the caspases pathway and the activation of JNK (c-Jun N-terminal kinase) and AKT kinases, associated with an increase in intracellular calcium, and independently of the activation of the PPARγ pathway [35]. In another study with 20 patient samples, higher PGE_2_ levels were observed in renal carcinoma samples compared with non-neoplastic renal parenchyma. However, these levels were not related to tumor size, Fuhrman grade, TNM (classification of malignant tumors and the extent of spread of cancer) stage, or histological subtype [77].

The most common of all upper urinary tract cancer are those found in the renal pelvis and renal calices. Cancer in the ureters makes up about a quarter of all upper urinary tract cancer. Tumors of the renal calices, renal pelvis and ureters start in the layer of tissue that lines the bladder and the upper urinary tract. Bladder cancer is a common solid tumor marked by high rates of recurrence, especially in non-muscle invasive disease. Inhibition of COX enzymes by NSAIDs results in reduced PGE_2_ levels, which is related with reductions in the bladder cancer [78]. Clinical trials using NSAIDs to prevent recurrence have had mixed results, but largely converge on issues with cardiotoxicity.

### 2.7. Prostaglandins in Nervous System Cancer

Nervous system cancer is the second leading type of cancer in children, after leukemia. Cancer of the nervous system involves tumors that form in one or more parts of the nervous system. Neuroblastoma, extracranial solid tumor of the sympathetic nervous system, mainly affects young children. Neuroblastoma is a cancer located in the adrenal medulla nerve cells of the body or other nervous system tissues such as the adrenal glands, around the spinal cord or in the abdomen. Cancer-associated fibroblasts is the main source of prostaglandin E2 in neuroblastoma contributing to angiogenesis, immunosuppression, and tumor growth.

Some authors believe targeting of mPGES-1 in cancer-associated fibroblasts will be an effective future therapeutic strategy in fighting neuroblastoma [79]. Several clinical and experimental studies have demonstrated that regular use of aspirin (ASA) correlates with a reduced risk of cancer and that the drug exerts direct anti-tumor effects. ASA could be used as an adjunctive therapeutic agent in the clinical management of neuroblastoma, although its effects appear to be mediated by a COX-independent mechanism involving an increase in p21 and underphosphorylated retinoblastoma (hypo-pRb1) protein levels [80]. MicroRNA-137 (miR-137) plays an important role in the development and progression of many types of human cancer. miR-137 was frequently down-regulated in retinoblastoma tissues. MiR-137 suppresses the proliferation and invasion of retinoblastoma cells by targeting COX-2/PGE_2_. miR-137 could be used as a potentially effective therapeutic target for the treatment of retinoblastoma [81].

Cancer of the nervous system might also affect the retina and it is called retinoblastoma or may affect the optic nerve and it is known as optic nerve glioma. Glioma, which is a cancer that afflicts the brain stem, is the most common type of cancer accounting for 45% of all brain cancer. There has been observed a direct and proportional relationship between the concentration of prostaglandin PGE_2_ and the degree of malignancy in glioma tumors. A higher concentration of PGE_2_ is related to greater cell proliferation and lower survival. This PG binds to EP2 and EP4 receptors, thereby activating PKA-II, which in turn activates CREB protein [82]. It has also been observed that PGE-EP4 binding stimulates the catabolic tryptophan 2-3 dioxygenase (TDO) enzyme pathway, which promotes immunosuppression by decreasing macrophages activation. The use of EP4 inhibitors decreases TDO activity [83]. Regarding prostaglandin PGD_2_, it has been shown that high doses slow tumor proliferation in glioma, but low concentrations seem to stimulate it [36]. On the other hand, it has been shown that the use of 15d-PGJ_2_ in human neoplastic glioma cells produces an increase in cell death, secondary to an increase in ROS production and activation of the caspases pathway [35].

### 2.8. Prostaglandins in Immune Cancer

Dendritic cells (DC) differentiate in the presence of determined factors or situations recognized as harmful, such as the onset of cancer. From the same progenitor, these cells differentiate into classical DC (cDC) or plasmacytoid DC (pDC) depending on which transcription factors are expressed. A study with mice showed that prostaglandin PGE_2_ synthesized in lymphoma cancer cells results in inhibition of ZBTB46 (zinc finger and BTB Domain Containing 46) factor expression (only expressed in cDC), thus preventing differentiation to cDC and favoring tumor expansion. In addition, the use of a COX-2 inhibitor, such as NS-398, resulted in functional cDC that reduced tumor burden [84].

Sperandio and coworkers demonstrated that treatment of multiple myeloma with 15d-PGJ_2_ produced a reduction in tumor cell proliferation without affecting non-neoplastic cells both in vitro and in mice tumors. A dose-independent increase in apoptosis was also observed in this study, which is probably related to the increased ROS concentration found in 15d-PGJ_2_-treated samples [85].

Acute lymphoblastic leukemia (ALL) develops in the bone marrow in the vicinity of stromal cells known to promote tumor development and treatment resistance. The COX inhibitor indomethacin prevents the ability of stromal cells to diminish p53-mediated killing of cocultured ALL cells in vitro, and in a xenograft animal model, possibly by blocking the production of PGE_2_. PGE_2_ released by bone marrow stromal cells might be a target for improved treatment of pediatric ALL. The indomethacin treatment increased the level of p53 in the leukemic cells, implying that COX inhibition might reduce progression of ALL by attenuating protective paracrine PGE_2_ signaling from bone marrow stroma to leukemic cells [86]. Glucocorticoid resistance remains a clinical challenge in pediatric acute lymphoblastic leukemia where response to glucocorticoids is a reliable prognostic indicator. cAMP signaling synergizes with dexamethasone to enhance cell death in glucocorticoid-resistant human T-ALL cells. The EP4 receptor expressed in T-ALL cells and PGE_2_ increases intracellular cAMP, potentiates glucocorticoid-induced gene expression, and sensitizes human T-ALL cells to dexamethasone in vitro and in vivo. [87].

On the other hand, an experiment performed with mice showed that the apoptotic effect of selenium (Se) supplementation in leukemia cells depends on the concentration of 15d-PGJ_2_., which were analyzed in serum samples. Se activates the PPARγ pathway in vitro and in vivo assays, which reduces STAT-5 (signal transducer and activator of transcription 5) levels and blocks the expression of the CITED2 (cAMP-responsive element-binding protein (CBP)) gene, one of the responsible genes for tumor quiescence [88].

### 2.9. Prostaglandins in Endocrine Tissue Cancer

Endocrine cancer is found in tissues of the endocrine system, which include the thyroid, adrenal, sexual glands, and pituitary glands. Pituitary adenomas are multifactorial intracranial neoplasms that impose a massive burden of morbidity on patients. A study demonstrated that COX-1 and COX-2 expression levels were increased in pituitary tumors from human patients, including non-functional pituitary adenomas, acromegaly, Cushing’s disease and prolactinoma compared with normal pituitary tissues. The level of PGE_2_ was consistent with COX enzymes expression in pituitary adenoma tumors compared with healthy pituitary tissue. These results may open new molecular targets for early diagnosis/follow up of pituitary tumor growth [89].

The cyclooxygenase-2 (COX-2)-prostaglandin E2 (PGE_2_) pathway with BRAF (serine/threonine-protein kinase B-Raf) mutation was shown to promote PGE_2_ synthesis. A study shows that COX-2 plays a key role in prognosis of Middle Eastern papillary thyroid carcinoma patients, especially in BRAF-mutated tumors. They suggest the potential therapeutic role of COX-2 inhibition in patients with BRAF-mutated papillary thyroid carcinoma [90]. It has been also observed that the use of prostaglandin 15d-PGJ_2_ on papillary thyroid cancer cells produces an increase in ROS due to the accumulation of intracellular iron, which triggers tumor cell apoptosis [36].

Regarding the male reproductive system, AKR1C3 protein appears to play a key role in androgen synthesis in prostate cancer cases. It has been shown that in AKR1C3^+^ cases, the concentration of 17β-HSD (17β-hydroxysteroid dehydrogenase) is high. This molecule can be blocked by androgen receptor (AR) antagonists, such as enzalutamide but in the final stage of this disease, this effect disappears due to the mutation of the target protein of the AR antagonist: TMPRSS2 (transmembrane protease, serine 2)-ERG (ETS (erythroblast transformation-specific)-related gene) [91]. Increased expression of AKR1C3 has also been associated with elevated concentrations of the prostaglandin PGF_2α_, which activates the MAPK pathway and inhibits the PPARγ pathway. All these events favor proliferation and tumor resistance of prostate cancer to radiotherapy. The use of indomethacin suppresses AKR1C3 and eliminates this resistance [92].

Prostaglandin PGE_2_ has been shown to increase the migration and invasiveness of the prostate cancer through activation of PI3K/AKT/mTOR and matriptase pathways. The use of COX-2 inhibitors such as CAY10404 and celecoxib has the opposite effect. In addition, prostaglandin 15d-PGJ_2_ acts as a tumor suppressor by inhibiting AR [93,94]. Finally, a cohort study with 50 patients showed that higher COX-2 levels were significantly associated with higher PSA (prostate-specific antigen), higher Gleason grade, worse prognosis, higher probability of relapse after treatment, and shorter survival time [95].

Endometrial and ovarian cancer represent most gynecological malignancies in developed countries. Personalized treatments for this cancer depend on identification of prognostic and predictive biomarkers that allow stratification of patients. A study evaluated the level of AKR1C3 in endometrial cancer and ovarian cancer and examined possible correlations between expression of AKR1C3 and other clinical and pathological data [96]. Thus, the expression of AKR1C3 was higher in endometrial cancer compared to ovarian cancer. In endometrial carcinoma, high AKR1C3 expression correlated with better overall survival and with disease-free survival. In patients with ovarian cancer, there was no correlation between AKR1C3 expression and overall and disease-free survival or response to chemotherapy. These results demonstrate that AKR1C3 is a potential prognostic biomarker for endometrial cancer. Another study showed that the binding of the prostaglandin PGF_2α_ to its receptor favors the proliferation and migration of tumor cells in patients with endometrial cancer [45]. On the other hand, prostaglandin PGJ_2_ has been shown to have inhibitory effects on endometrial tumor cell proliferation [91].

It has been established that an increase in prostaglandin PGE_2_ provides ovarian tumor cells with increased resistance to chemotherapy [97]. There is a known link between low expression of RGS10 (regulator of G-protein signaling 10), a G-protein modulator of prostaglandin action, and increased resistance to chemotherapy treatment in ovarian cancer. The loss of functional RGS10 results in an increase in COX-2 concentration and thus an increase in prostaglandin PGE_2_ synthesis, which leads to the conclusion that RGS10 inhibits COX-2. However, this inhibition seems to be independent of its action on G proteins, since when inhibitors of this protein were used, no increase in the concentration of inflammatory markers was observed, so that the specific mechanism of inhibition has not yet been fully determined.

Even though in general cancer cases an increase in COX-2 levels is more common, it is very interesting that in the case of ovarian cancer a higher concentration of COX-1 is observed, which is why it has been suggested as a biomarker for early diagnosis. It has been proposed to use COX-1 inhibitors such as [18F]-fluorin and [18F]-P6 as tracers to detect ovarian cancer when performing PET (positron emission tomography). Furthermore, it has been shown that the treatment effect is increased with placitaxol combined with a selective COX-1 inhibitor (SC-560). It can be concluded that COX-1 inhibitors favor chemosensitivity of ovarian cancer [98].

To identify biomarkers that could predict response or lack of response to conventional chemotherapy at the time of diagnosis of high grade serous ovarian carcinoma, a study showed that evaluation of PGD_2_ is an independent marker of good prognosis in this carcinoma [99]. On the other hand, infection by human papillomavirus (HPV) serotype 16 is associated with an increase in COX-2 synthesis, which may be related to cases of cancer associated with these infections [100].

### 2.10. Prostaglandins in Other Cancer

COX-2 is activated in response to inflammatory stimuli, and it can mediate its tumorigenic effect through various mechanisms, such as inducing cell proliferation, inhibition of apoptosis, and suppressing the host’s immune response. Furthermore, COX-2 can induce the production of vascular endothelial growth factors, hence, promoting angiogenesis. The ability of COX-2 inhibitors to selectively restrict the proliferation of tumor cells and mediating apoptosis provides promising therapeutic targets for cancer patients. It is believed that COX2 can promote the development and progression of head and neck cancer. Therefore, COX2 inhibitors could be promising therapeutic weapons to fight the cancer [101].

EMT and angiogenesis are crucial events for development of aggressive and often fatal oral squamous cell carcinomas. Both processes promote cancer progression and metastasis development, but while the former induces the loss of E-cadherin expression; the latter produces blood vessel neoformation and contribute to cell growth, tumor mass development, and dissemination. COX-2 decreases the expression of E-cadherin and leads to phenotypic changes in epithelial cells enhancing their carcinogenic potential. This study was performed by a tissue microarray of oral squamous cell carcinomas from human patients [102].

An experiment with 180 mice demonstrated the potential of an EP-4 receptor inhibitor such as GW627368X to reduce PGE_2_ expression in sarcomas treatment [103]. This treatment resulted in reduced tumor volume and weight, and induction of apoptosis with increased levels of BAX (BCL-2-like protein 4) and AIF (apoptosis-inducing factor) along with low levels of *Mcl-1* (induced myeloid leukemia cell differentiation protein) and *Bcl-2* (apoptosis regulator). Moreover, this treatment did not cause systemic toxicity, immunosuppression, or behavioral changes.

Fibroblasts induce a high expression of COX-2, which leads to increased synthesis of PGs that promote the secretion of fibroblast growth factor (FGF) and vascular endothelial growth factor (VEGF), thus generating a continuous positive feedback. One study shows that the use of DAPS (2,5-diacetyloxyphenylsulfonate), an inhibitor of FGF and VEGF secretion, stops angiogenesis and tumor cell expansion in mice [104].

### 2.11. Prostaglandins in Tumor Microenvironment and Metastasis

The main reason cancer is such a serious illness is because of its ability to spread in the body. Cancer cells can spread locally by moving within the surrounding normal tissue. Cancer can also spread regionally, to nearby lymph nodes, tissues, or organs, and it can also spread to distant parts of the body. When this happens, it is called metastatic cancer or metastasis. For many types of cancer, this is called stage IV cancer. The transformation of a tumor cell into a metastatic tumor cell probably involves transient or permanent genetic changes, which determine the expression of molecules with actions that favor or protect the mechanisms necessary for metastasis. The most frequent locations of metastases are the organs most irrigated by blood, such as the brain, lungs, liver, bones, and adrenal glands. There is also a tendency for certain tumors to spread in certain organs. For example, prostate cancer tends to spread through the bones. Likewise, colon cancer does so in the liver and stomach cancer in the ovaries in the case of women. The cancers that metastasize the most are the most frequent cancers such as breast cancer, lung cancer, melanoma, and colorectal cancer.

Tumorigenesis is a multistep biological process and many studies have been focused on the critical role of the tumor microenvironment (TME). The tumor microenvironment plays a major role in the ability of the tumor cells to undergo metastasis. Tumor cells along with a small proportion of cancer stem cells exist in a stromal microenvironment consisting of vasculature, cancer-associated fibroblasts, immune cells, and extracellular components.

A major player of tumors gaining metastatic property is the inflammatory protein COX-2. Several tumors show upregulation of this protein, which has been implicated in mediating metastasis in various cancer types such as of colon, breast, and lung. COX2 has been closely linked to the occurrence, progression, and prognosis of cancer in tumor microenvironment [105]. It has also been shown that an abnormally high expression of the COX-2 enzyme leads to an increase in the concentration of the different PG subtypes, which favor or hinder different aspects of carcinogenesis. 15-Deoxy-Δ12,14-prostaglandin J2 (15d-PGJ_2)_ is a prostaglandin whose effects depend on its concentration and the cell type in which it is found. Its effects include a decrease in angiogenesis and favoring of the apoptosis process. Prostaglandin 15d-PGJ_2_ produces an increase in ROS through mitochondrial dysfunction, activation of NADPH (Nicotinamide adenine dinucleotide phosphate) oxidase and JNK, and inhibition of AKT, which favors tumor apoptosis through the TRAIL (TNF-related apoptosis-inducing ligand) pathway associated with an increase in the concentration of the DR5 receptor, through a mechanism independent of the PPARγ pathway [36]. This activates PI3K-Akt signaling in human breast cancer cells through covalent modification of the tumor suppressor PTEN at cysteine 136 [46]. One study has shown that PGE_2_ favors the angiogenesis process for the supply of oxygen and nutrients necessary for tumor progression. The use of 3-(3-methylthiophen-2-yl)-5-(3,4,5-trimethoxyphenyl) isoxazole (2b) showed significant inhibitory activity toward COX-2 and showed a good inhibition of tumor growth, peritoneal angiogenesis, and ascites formation in Ehrlich ascites carcinoma (EAC) cell mouse model [106].

Increased levels of ATP in the tumor microenvironment in response to cell death mediated by chemotherapeutic agents such as doxorubicin leads to increased COX-2 expression, which, in turn, affords migratory and invasive properties to the tumor. Anti-inflammatory drugs against ATP receptors is a potential opportunity to be explored as cancer therapeutics [107]. Recent epidemiological and clinical studies strongly support the assertion that vitamin D supplementation is associated with reduced cancer risk and favorable prognosis. Experimental results suggest that vitamin D not only suppresses cancer cells, but also regulates tumor microenvironment to facilitate tumor repression [108]. Another study with mice showed that an increased intake of omega-3 and omega-6 polyunsaturated fatty acids reduced tumor growth in the melanoma microenvironment, along with a lower concentration of prostaglandins PGE_2_ and PGE_3_ [109].

The hypothesis that the anti-inflammatory activity of NSAIDs should be based on COX inhibition is well established in multiple experimental models, and NSAIDs are one of the therapeutic tools used in the current clinical fight against cancer, combined with other therapies. Thus, COX-2 inhibitors can reduce inflammatory factors thereby regulating macrophage recruitment for activating the antitumor immune microenvironment; downregulating vascular endothelial growth factor (VEGF) to inhibit tumor angiogenesis; and inhibiting the PI3K/Akt signaling pathway to induce tumor cell apoptosis. However, further in-depth investigation of these drug is needed to maximize antitumor efficacy and minimize the side effects [110,111].

The molecular and cellular mechanisms that cancer cells use to spread to other organs are diverse and complex. Thus, activation of PPARγ serves as a key factor in the proliferation and invasion of breast cancer cells and it is a potential therapeutic target for breast cancer. Heme oxygenase-1 (HO-1) is induced and overexpressed in various types of cancer and is associated with features of tumor aggressiveness. Recent studies have shown that HO-1 is a major downstream target of PPARγ. Jang and coworkers suggest that 15d-PGJ_2_ inhibits MMP9 expression and invasion of breast cancer cells by means of a heme oxygenase-1-dependent mechanism. Therefore, PPARγ/HO-1 signaling pathway inhibition may be beneficial for prevention and treatment of breast cancer and its metastasis [112].

The host stromal mPGES-1 is involved in the accumulation of myeloid-derived suppressor cells (MDSCs) in metastasized lungs of prostate cancer in mice. mPGES-1 enhances tumor metastasis in mice by inducing accumulation of BM (bone marrow)-derived MDSCs. Selective mPGES-1 inhibitors might, therefore, represent valuable therapeutic tools for the suppression of this tumor metastasis [113]. Prostaglandin 15d-PGJ_2_ has the ability to decrease breast and renal cancer metastasis, since it produces a reduction in the synthesis of extracellular matrix proteins (MMP), such as MMP-2 and MMP-9, which minimizes the invasiveness of these tumor cells [35]. Also, less melanoma metastasis has been demonstrated in mice with elevated concentrations of PGD_2_ [36]. By contrast, another study with patients showed that a higher concentration of PGE_2_ is significantly related to a higher expression of metalloprotease-1 (MMP1), which facilitates the transfer of cancer cells through the hematoencephalic membrane, and with it, the appearance of brain metastasis of breast cancer [114]. Similarly, prostacyclin (PGI_2_) has been shown to have great relevance as a tool to stop the development of tumor metastases [115]. Finally, the use of intraoperative injections of Etodolac (a NSAID) and propranolol in mice has been shown to reduce the number of colon cancer liver metastases by activating NK cells. However, neither compound separately obtained significant results [116].

Cervical cancer metastasis results in poor prognosis and increased mortality, which is not separated from inflammatory reactions accumulated by PGE_2_. As a specific G-protein coupled PGE_2_ receptor, EP3 is demonstrated as a negative prognosticator of cervical malignancy. EP3 signaling pathway might facilitate the migration of cervical cancer cells through modulating uPAR (urokinase plasminogen activator surface receptor) expression. Therefore, EP3 and uPAR could represent novel therapeutic targets in the treatment of cervical cancer in advantaged stages [117].

The ability of the immune system to recognize and eliminate tumor cells is called immune surveillance. One of the functions of the immune system is to recognize and destroy tumor cells before they grow and form a cancer, as well as to eliminate tumors that have already formed. At first, the results of some experiments questioned the importance of immune surveillance, but today it is clear that the immune system reacts to many tumors. This knowledge has led to the immune response being seen as another therapeutic weapon in the treatment of cancer. MDSCs include immature monocytic (M-MDSC) and granulocytic (PMN-MDSC) cells that share the ability to suppress adaptive immunity and to hinder the effectiveness of anticancer treatments. In response to IFNγ (interferon gamma), MDSCs release the tumor-promoting and immunosuppressive molecule nitric oxide (NO), whereas macrophages largely express antitumor properties. A study indicates that inhibition of the PGE_2_/p50/NO axis prevents MDSC-suppressive functions and restores the efficacy of anticancer immunotherapy [118].

Immune checkpoint inhibitors improve survival outcomes in metastatic melanoma and non-small cell lung cancer. Preclinical evidence suggests that overexpression of cyclo-oxygenase-2 (COX2) in tumors facilitates immune evasion through prostaglandin E2 production and that COX inhibition synergizes with immune checkpoint inhibitors to promote antitumor T-cell activation. COXi inhibitor used concurrently with immune checkpoint inhibitors in human patients significantly associated with longer time-to-progression and improved objective response rate at 6 months in patients with metastatic melanoma and non-small cell lung cancer compared with Immune checkpoint inhibitors alone [119]. Furthermore, COXi inhibitor use appears to reverse the negative prognostic effect of a high neutrophil-lymphocyte ratio by prolonging time-to-progression in patients with melanoma.

Resistance to immunotherapy is one of the biggest problems of current oncotherapeutics. White T cell abundance is essential for tumor responsiveness to immunotherapy, factors that define the T cell inflamed tumor microenvironment are not fully understood. Markosyan and coworkers demonstrated in mice that tumor cell-intrinsic EPHA2 suppresses anti-tumor immunity by regulating PTGS2 (COX-2) in pancreatic adenocarcinoma [120]. Polymorphonuclear myeloid-derived suppressor cells (PMN-MDSCs) are pathologically activated neutrophils that are crucial for the regulation of immune responses in cancer. These cells contribute to the failure of cancer therapies and are associated with poor clinical outcomes. A work demonstrated that mouse and human PMN-MDSCs exclusively upregulate fatty acid transport protein 2 (FATP2). Thus, FATP2 mediates the acquisition of immunosuppressive activity by PMN-MDSCs and represents a target to inhibit the functions of PMN-MDSCs selectively and to improve the efficiency of cancer therapy [121]. The initiation of an intestinal tumor is a probabilistic process that depends on the competition between mutant and normal epithelial stem cells in crypts. Intestinal stem cells are closely associated with a diverse but poorly characterized network of mesenchymal cell types. One study performed in humans and mice demonstrates that initiation of colorectal cancer is orchestrated by the mesenchymal niche and reveals a mechanism by which rare pericryptal *Ptgs2*-expressing fibroblasts exert paracrine control over tumor-initiating stem cells via the druggable PGE_2_-PTGER4-YAP signaling axis [122].

Table 2 summarizes some of the data observed in the different studies analyzed, which could help to plan new research.

## 3. Conclusions

According to the data revised in this work, the involvement and importance of PGs in the development of different neoplasms is more than evident. The effects of prostaglandin in cancer depends on multiple factors, such as the target damaged tissue, the plasma concentration of prostaglandins and their subtypes, and the presence of genetic mutations and the different intracellular signaling pathways involved, will tip the balance either towards tumor regression or cancer expansion.

Prostaglandins of the PGD_2_ type are clearly related to better survival expectations, since they can hinder tumor progression. In all the evidence reviewed, it can be observed that a higher concentration of the prostaglandin PGD_2_ is associated with a greater difficulty for the tumor to progress, either as a primary tumor or as metastasis, and with increased apoptosis of cancer cells. However, a low concentration of this prostaglandin does not protect against tumor development. Therefore, PGD_2_ can be considered as a protective factor against cancer and a marker of good prognosis.

PGE_2_ and PGF_2α_ prostaglandins are related to greater tumor progression and aggressiveness, also associated with a worse functioning of the immune system. With respect to PGE_2_, all the references found seem to point to it as a prostaglandin that facilitates tumor development by activating different signaling pathways. The action of this prostaglandin results in greater vascularization, greater ease of metastasis, and less maturation of immune system cells capable of halting its advance, like macrophages polarization and activation, although it sensitizes human T-ALL cells to dexamethasone. Therefore, it could be said that a high amount of PGE_2_ is a risk factor to be considered in tumor origin and progression. In fact, some of the authors propose using the measurement of its levels as a biomarker of neoplastic evolution. Everything seems to indicate that the prostaglandins PGF_2α_ is one of the factors responsible for the resistance of different types of cancer to some of the therapeutic options, such as the use of ASA or chemotherapy. In addition, this prostaglandin has been related to a greater capacity for cancer migration and invasion and prevents tumor apoptosis. Therefore, elevated PGF_2α_ concentration could be considered a risk factor for neoplastic progression. Also, PGF_2β_ facilitates tumor progression in some neoplasms such as colorectal cancer.

Prostaglandins PGI_2_ and PGJ_2_ are generally related with the arrest of cancer development, but since works point them out as possible inducing factors of neoplasia, further research is necessary to clarify the contradictory data. Thus, there is some controversy in the implication of prostaglandin PGI_2_ in cancer. Most scientific works show data that may lead to the assumption that it acts as a protective factor against neoplastic development. Thus, high concentrations of PGI_2_ are related to a lower number of metastases of any type of cancer and a higher 5-year survival in squamous epithelial cancer by inhibiting lung tumor growth. However, one research work demonstrates a significant relationship between a high concentration of this PG and a lower survival in breast cancer. Therefore, determining the role of this PG in the development of cancer is risky, so it would be advisable to study its involvement in different types of cancer. As with prostaglandin PGI_2_, the data on the involvement of prostaglandin PGJ_2_ in cancer are also contradictory. Most of the works suggests that PGJ_2_ has the capacity to inhibit tumor expansion, as it seems to facilitate cancer cell apoptosis and hinder angiogenesis, proliferation, and invasion. However, some other researchers propose that PGJ_2_ may promote tumor progression by inhibiting the tumor suppressor PTEN activity in breast cancer.

In addition to the evidence found about the influence of the different subtypes of PGs on the distinct types of cancer, it is necessary to make special mention of the function of AKR1C3 protein that can determine the intensity of the effects of these lipid molecules on tumor development. The presence of this protein has been related to a shift from the synthesis of “potentially protective PGs”, such as PGD_2_, towards the synthesis of “potentially carcinogenic PGs”, such as PGF_2α_. This shift leads to increased tumor vascularization and increased resistance to different therapeutic tools. For this reason, the AKR1C3 protein is proposed as a possible and interesting therapeutic target. In fact, in some of the studies reviewed, inhibitors of this enzyme are used, in which PGF_2α_ levels are normalized and tumor expansion is slowed down.

The use of different NSAIDs, especially selective COX-2 inhibitors, may have a crucial role in the therapeutic strategy to end the origin and progression of neoplasms, either as a prophylactic treatment in the population at risk, or as an adjuvant treatment to chemotherapy or radiotherapy. Moreover, the analysis of the intracellular signaling pathways involved in cancer and related to the action of prostaglandins has revealed multiple therapeutic targets in addition to COX1/2 enzymes. Therefore, more detailed studies should be carried out to obtain better therapeutic tools by combining treatments against cancer.

Given the great relevance of PGs in terms of the outcome of neoplasms, it would be interesting to favor, by means of different genetic, biochemical, and molecular tools, the reduction of the concentration of “potentially carcinogenic PGs”, such as PGE_2_ and PGF_2α_, and to increase the concentration of “potentially protective PGs”, such as PGD_2_, PGI_2_ and PGJ_2_. Therefore, both the inhibition of the expression of the genes that code for COX enzymes, and the use of specific inhibitors of these enzymes are good therapeutic strategies against cancer, since they would reduce the concentrations of the different types of PGs.

On the other hand, the scientific studies reviewed also consider other therapeutic possibilities, such as the use of PG receptor inhibitors. Despite the high concentrations of PGs that may be present in plasma in a neoplastic process, the inhibition of their receptors would avoid the binding of PGs to their receptors and PGs would not be able to produce any effect on the nearby cells, with the result that the PGs would be destroyed without having any repercussions. In addition, as described above, there are many signaling pathways that interact with PGs. These interactions increase the number of possible therapeutic targets, since it would be possible to block these connections at more points, at the genetic, molecular, or cellular levels. Some examples of these potential targets analyzed are ZIP5, mPGES-1, FGF, VEGF, XRCC5, PPARγ, AKT, ERK1/2 MAPK, NFκB, PTEN, or p300, among many others. 

Finally, as described above, the studies should be also focused on the critical role of the tumor microenvironment (TME) as a promoter of the ability of the tumor cells to undergo EMT and metastasis. One of the major players in metastatic tumors is the inflammatory protein COX-2 upregulated in several tumors, which lead to the production of PGs and the development of the different aspects of carcinogenesis. As mentioned through this revision, selective COX-2 inhibitors are preferred over an inhibitor of both isoforms to avoid the side effects of COX-1 suppression. COX-2 selective inhibitors produce a reduction of PGs together with the inhibition of tumor progression, either in extension or in the number of tumors. The hypothesis that the anti-inflammatory activity of NSAIDs should be based on COX inhibition is well established, and NSAIDs are one of the therapeutic tools used in current clinical fight to cancers, combined with other therapies. Another factor to consider in the fight against cancer is the immune surveillance or the ability of the immune system to recognize and eliminate tumor cells, which has been seen as another therapeutic weapon in the treatment of cancer.

It is important to clarify that data presented in this review, about the involvement of prostaglandins in the different types of cancer, are observations made in vitro assays, by using cell lines, in ex vivo assays, by using samples of tumorigenic tissues from animals or humans, or in vivo experiments, made in animals, but few molecular and cellular data have been confirmed in humans. Therefore, more detailed studies should be carried out in humans to confirm the molecular and cellular events observed in animals or cell lines, and to determine the appropriate concentrations of prostaglandins and their inhibitors to use in human patients in order to develop safe single or polytherapy tools against cancer, avoiding unwanted side effects as much as possible.

## Figures and Tables

**Table 1 cells-10-01487-t001:** Summary of prostaglandin (PG) general functions.

Biological System	PG Mediator	Physiological Effect
Digestive system	PGE_2_, PGI_2_	Reduction of acid secretion; Increase of mucous secretion
	PGE_2_	Longitudinal smooth muscle contraction; Circulatory smooth muscle contraction
Respiratory system	PGI_2_, PGE_2_	Bronchodilator
	PGH_2_, PGF_2_α	Bronchoconstriction
Cardiovascular system	PGE_2_, PGI_2_	Arterial vasodilation
	PGF_2_α	Inhibition of platelet adhesion and leukocyte aggregation
Renal system	PGI_2_, PGE_2_	Medullary blood flow, pressure diuresis
	PGI_2_, PGE_2_	Renin release
	PGE_2_	Natriuresis, diuresis
Immune system	PGE_2_, PGI_2_	Inhibition of proliferation and activation of T and B lymphocytes
Central nervous system	PGE_2_	Inflammation
	PGD_2_, PGI_2_	Induction of sleep
Female reproductive system	PGE_2_, PGI_2_, PGF_2_α	Ovulation, implantation, endometrial contraction, and synergism with oxytocin
Male reproductive system	PGE_1_, PGE_2_, PGE_3_, PGF_2_α	Fertility

**Table 2 cells-10-01487-t002:** Summary of the role of prostaglandins in cancer. In red: cellular, molecular factors or treatments that promote the development of cancer. In blue: cellular, molecular factors or treatments that hinder tumor progression.

Tissue/Organ	Type of Cancer and Scenario	Cellular-Molecular Factors/Therapeutics	Molecular/Cellular Effect	Physiological/Pathological Impact
**Skin**	Squamous	AKR1C3^+^	PGD_2_ decrease.	Promotion of neovascularization
			PGF_2α_ increase	
		15d-PGJ_2_	Inhibition of STAT-3 pathway	Reduction of cell growth
		PGI_2_ increases		Higher 5-year survival rate
		miR-31-5p increases	ACOX-1 decrease.	Increase of tumor migration and invasion
			PGE_2_ increase	
		PGE_2_	PGE_2_ increase, larger stage	Possible biomarker of progression?
	Non-melanoma	Apigenin	COX-2 and PGE_1_-EP1/EP2 decrease	Inhibition of neoplastic progression
		Piroxicam	COX-2 decrease	Useful in prevention
	Melanoma	PGF2_α_	Blocking AAS action	Prevention of tumor apoptosis
		PGF2α antagonist	Inhibition of ASA blockade	Promotion of tumor apoptosis
		Topical Melanotan II	Inhibition of COX-2 expression and PGE_2_ production	Inhibition of the migration, invasion, and colony-forming capability
**Bones**	Osteosarcoma	SND1	Increase of PGE_2_	Antitumor strategy using COX2 inhibitors.
				Potential biomarker of the therapeutic strategies
		PGE_2_	>EP1 pathway	Increase of proliferation and decrease of apoptosis of cells
		17-PT-PGE_2_		
**Lungs**	CRTC1+/LKB1 cancer	CRTC1^+^/LKB1^−^	Activation of cAMP/CREB y PGE_2_	Promotion of tumor development
		Niflumic acid (NS-398)	PGE_2_ decrease	Hindrance of tumor development
	Non-small cell cancer	miR-574-5p	Decrease of CUGBP1 and increase of mPGES-1 and PGE_2_	Promotion of tumor development
		15d-PGJ_2_	Increase of ROS and activation of caspases	Increase of apoptosis
		PGD_2_		
		mPGES-1 inhibitors	PGE_2_ decrease	Cancelation of miR-574-5p effects
	MHC-II^+^ lung cancer	PGIs	Increase of T-CD4^+^ lymphocytes	Inhibition of tumor growth
	Primary lung tumors	ACSL3	Increase of LPIAT1 activity	Prediction of poor patient survival.
				Anchorage-independent growth
	Lung squamous cell carcinoma	PGE_2_	Activation of TNF-alpha-TRAF2-MMP-9	Progression of lung cancer
		TNF-alpha		
	General lung cancer	NSAIDs	Inhibition of COX enzymes	Smaller tumor size and fewer metastasis.
Mammary gland	Breast cancer	AKR1C3^+^	PGF_2α_- FP and Ki-67 increase	Increase of cell proliferation
		AKR1C3 inhibitor	PGF_2α_ decrease	Reduction of cell expansion
		PGF2_α_-FP increase	Activation of ERK1/2-MAPK pathway and activation of NF-κB	Increase of resistance to QT
		FP inhibitor	Inhibition of ERK1/2-MAPK pathway	Reduction of resistance to QT
		NF-kB inhibitor	Inhibition of NF-κB factor	
		15d-PGJ_2_	Activation of AKT-AP-1 pathway	Promotion of tumor expansion.
		15d-PGJ_2_	Up-regulation of Snail and CXCL8 expression	Epithelial-to-mesenchymal transition (EMT).
				Tumor-stroma interaction
		8-iso-PGF_2α_	Serum non-invasive marker	Oxidative stress and subsequent damaging of DNA
		C136S-PTEN (mutated)	Not affected by 15d-PGJ_2_	Resistant?
		DGLA *	Activation of caspases, PARP and COX-2	Decrease of tumor migration and invasion. Greater efficacy of treatment with 5-fluouracil.
		PGE_2_-EP2 increase	CD80 decrease on macrophages	Reduced macrophage polarization
		PGESm-1 Knock out	PGE_2_-EP2 decrease and CD80 increase	Normal macrophage polarization
		PGI_2_ increase		Shorter survival time
		Ibuprofen	PGE_2_ decrease	Less tumor volume (dose-dependent), more mature macrophages, more CD-45^+^ T-lymphocytes, and fewer immature monocytes
		Propanolol + Etodolac (Peri-QX)	Inhibition of STAT and EGR3 pathways	Less tumor dissemination, more NK lymphocytes, more B cells, fewer monocytes, and less IL-6.
**Liver**	HCC	Increase of stellate cells	COX2-PGE_2_-EP4 increase	Fewer regulatory T-lymphocytes, more MDSCs
		SC-236 (COX-2 inhibitor)	Stellate cells ** decrease	Stop the spread of cancer
		AH23848 (EP4 inhibitor)		
		PGE_2_	EP4-G-Adenylate increase and activation of cyclase-cAMP-kinase A-CREB pathway and oncogene MYC	Facilitation of tumor expansion
	Hepatitis B virus (HBV)-related hepatocellular carcinoma (HCC).	2,5-dimethylcelecoxib (DMC). PD-L1	Inhibition of microsomal prostaglandin E synthase-1 (mPGES-1)/PGE_2_ production	DMC combined with atezolizumab: more antitumor effect and stronger blockage of immunosuppression effect on PD-L1
**Digestive system**	Esophageal squamous cancer	ZIP5 inhibitor	Cyclin D1 decrease. COX-2 increase	Metastasis reduction
	Gastric adenocarcinomas	*H. pylori*	COX-2 increase	Promotion of the onset of neoplasia
		NSAIDs	Inhibition of COX	Effective prophylaxis
		PGD_2_	PPARγ decrease	Slower growth
		15-PGDH	FOXP3	Anti-tumor immunity
	Adenoma	EP4 inhibitors	Inhibition of PI3K-AKT-mTOR and ERK1/2-MAPK pathways	Reduction of the number of new adenomas
		ASA as prophylactic (75–325 mg)		Fewer adenomas and lower mortality
		Sulindac (15-PGDH knock-out)		Fewer new adenomas and more inflammatory lesions
	Colorrectal cancer	COX2	Inhibition of COX2	Prevention of carcinogenesis. Increase in the survival rate.
				Risk of cardiovascular complications with prolonged treatment
		Targeting the TME	Downstream molecules of PGE_2_ signaling	Promising approach
		PGF_2α_		Increased migration and invasion
		15d-PGJ_2_	MYC modulation and telomerase inhibition	Increased rate of cell death
		CRTC1	Increase of CREB/AP-1, COX-2 and aaPGE_2_	Promotion of tumor development
		IP6	Decrease of COX-2 and PGE_2_	Hindrance of tumor development
		NSAIDs		Reduction of tumor mass and metastasis
		AAS (Stage III)		Lower mortality and relapses
		AAS + Statins	PG decrease	
		PGF_2β_	Increase of EGR1 factor and prostaglandin synthase E enzyme	Promotion of tumor progression
		PGF_2β_ inhibitor	Decrease of EGR1 factor and prostaglandin synthase E enzyme	Hindrance of tumor progression
		Tumor suppressor Knock-out. 15-PGDH		Increased resistance to ASA and celecoxib
		PGM increase	Elevated levels: patients already diagnosed > patients with multiple adenomas > healthy controls.	Early diagnostic marker?
		XRCC5 protein	p300 and COX-2 increase	Promotion of tumor progression
		p300 inhibitor	COX-2 decrease	Hindrance of tumor progression
**Pancreas**	Pancreatic cancer	AAS		Not very useful, since they do not express COX-1.
		Celecoxib	COX-2 decrease	Possible adjuvant treatment for cisplatin + gemcitabine?
		Vitamin D3 analogues: calcipotriol	PD-L1 upregulation	Decreased cancer-associated fibroblasts proliferation and migration. Reduced release of PGE2.
**Kidney**	Renal cancer	15d-PGJ_2_	Activation of caspases, and JNK and AKT kinases. Intracellular	Promotion of apoptosis
			[Ca^2+^] increase	
		COX-1 increase		Higher degree of malignancy
		PGE_2_ increase		Not related to tumor size, Fuhrman grade, TNM stage or histological subtype.
		Cadmium	Activation of cAMP/PKA II-COX2 pathway and N-Catherin expression	Mediated cell migration and invasion
**Urinary system**	Bladder cancer			
**Nervous system**	Glioma	PGE_2_-EP2	PKA-II and CREB increase	Increase of proliferation and decrease of survival
		PGE_2_-EP4	TDO decrease	Reduction of macrophage activation
		PGD_2_ increases		Reduction of tumor proliferation
		PGD_2_ decreases		Increase of tumor proliferation
		15d-PGJ_2_	ROS and caspases increase	Increase of cell death
	Neuroblastoma	ASA	COX-independent mechanism involving an increase in p21 and underphosphorylated hypo-pRb1.	Adjunctive therapeutic agent
	Retinoblastoma	MicroRNA-137	Inhibition of COX-2/PGE_2_	Suppression of proliferation and invasion
**Immune system**	Multiple Myeloma	15d-PGJ_2_	ROS increase	Increase of angiogenesis and promotion of apoptosis
			via PPARγ decrease	
		Increased intake of omega-3 and omega-6 polyunsaturated fatty acids	PGE_2_ y PGE_3_ decrease	Reduction of tumor growth
	Acute lymphoblastic leukemia (ALL)	Indomethacin	Avoid the stromal cells diminished p53-mediated killing. Blockage of the production of PGE_2_	Reduction of progression of ALL
		EP4 receptor	Increase of intracellular cAMP	Sensitizes human T-ALL cells to dexamethasone
		PGE_2_		
	General Leukemia	Selenium supplements	Activation of PPARγ. Inhibition of STAT-5 and CITED2	Apoptotic effect
		15d-PGJ_2_	Increase of ROS-NADPH oxidase. Activation TRAIL-JNK. Inhibition of AKT	
	Lymphomas	PGE_2_	Factor ZBTB46 decrease	Prevention of differentiation to cDC
		NS-398	PGE_2_ decrease and cDC increase	Tumor burden reduction
**Endocrine tissues**	Pituitary adenomas	COX1/2 PGE_2_		Promotion of tumor progression
	Papillary thyroid cancer	15d-PGJ_2_	[Fe^2+^] intracellular and ROS increase	Promotion of tumor apoptosis
		COX2 and PGE_2_	BRAF-mutated tumors promote PGE_2_ synthesis	Promotion of tumor progression
	Prostate cancer	AKR1C3^+^	Increase of PGF_2α_ and activation of MAPK pathway.	Increase of proliferation and resistance to radiation therapy
		17β-HSD	Inhibition of PPARγ	

		Androgen receptor antagonists, such as enzalutamide	Blockage of 17β-HSD	Indomethacin suppresses AKR1C3 and eliminates resistance
		PGE_2_-EP1/EP2	Activation of PI3K/AKT/mTOR and matriptase pathways	Increase of migration and invasion
		CAY10404 and celecoxib	PGE_2_ decrease. Inhibition of PI3K/AKT/mTOR and matriptase pathways	Decrease of migration and invasion
		15d-PGJ_2_	Inhibition of AR	Tumor suppressor
		COX-2 increase	PSA and Gleason increase	Poorer prognosis, more relapses, and poorer survival.
	Endometrial cancer	PGF2_α_		More proliferation and migration
		AKR1C3^+^		Better overall survival Prognostic biomarker
		PGJ_2_		Reduction of proliferation
	Ovary cancer	RGS10 decrease	COX-2 and PGE_2_ increase	More resistance to chemotherapy
		COX-1 *** increase		Early diagnostic biomarker?
		COX-1 inhibitors ([^18^F]-Fluorine y [^18^F]-P6)		Trackers when performing a PET scan?
		SC-560		Increased chemosensitivity
	Serous ovarian carcinoma	PGD_2_		Marker of good prognosis
	HPV serotype 16 infection		COX-2 increase	Related to the onset of cancer?
	Cervical cancer	PGE_2_ receptor, EP3	Modulation of uPAR expression	Negative prognosticator of cervical malignancy
**Other tissues**	Sarcoma	GW627368X (EP4 inhibitor)	BAX and AIF increase.	Reduction of tumor volume and weight. Induction of apoptosis
			MCL-1, BCL-2 and PGE_2_ decrease	
		Fibroblasts	COX-2, PGE_2_, FGF and VEGF increase	Increase of angiogenesis and tumor spread
		DAPS	FGF and VEGF decrease	Decrease of angiogenesis and tumor spread
	Head and neck cancers	COX-2	Various mechanisms	Protumorigenic effect. COX-2 selective inhibitors
	Oral squamous carcinomas	COX-2	Loss of E-cadherin expression	EMT and angiogenesis
**TME/Metastasis/Immune surveillance**	All	COX-2. PGE_2_, ATP	Increased angiogenesis	Supply of O2 and nutrients.
		Vitamin D	Decrease of cancer risk and favorable prognosis	Sensitivity to NSIAIDs targeting PGE_2_
	Breast cancer	CXCL8	Activator of fibroblasts,	Tumor-stroma interaction in TME
	Liver cancer	2,5-dimethylcelecoxib (DMC)	Promotion of HBV-related HCC immune TME	Combined immunotherapy with DMC and atezolizumab
	Pancreatic adenocarcinoma	EPHA2	PTGS2 (COX-2)	Suppression of anti-tumor immunity
	PMN-MDSCs		Increase of FATP2	Immunosuppressive activity
	Intestinal tumor	Mesenchymal niche	PGE_2_-PTGER4-YAP signaling axis	Initiation of colorectal cancer
	Breast cancer	PGE_2_	MMP1 increase	More brain metastases
		PGI_2_		Stops its development
		15d-PGJ_2_	Inhibition of MMP9 through PPARγ/HO-1 signaling pathway	Prevention and treatment of breast cancer and its metastasis
	Prostate cancer	mPGES-1	Accumulation of BM-MDSCs in lungs	Use of Selective mPGES-1 inhibitors
	Renal cancer	15d-PGJ_2_	MMP decrease	Reduction of invasiveness
	Melanoma	PGD_2_		Lower number of metastasis
	Melanoma and non-small cell lung cancer	COX1/2 inhibitors		Lower number of metastasis
	Colorectal cancer	Etodolac + Propanolol	Lymphocytes NK increase	Reduction of tumor progression
	Immune System	Immune surveillance	Immune system reacts to many tumors.	Therapeutic weapon to eliminate tumor cells
	M-MDSC	Increase of NO	Suppression of adaptive immunity.	Inhibition of the PGE_2_/p50/NO

* After inhibiting the enzyme delta-5-desaturase (whose function is to convert DGLA into arachidonic acid). ** Only in vitro assays. *** Although in cancer COX2 is usually increased, in this case there is a very striking increase in COX-1.

## Data Availability

Data sharing not applicable.

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
