# Peer review of "The Role of Prostaglandins in Different Types of Cancer"

_cells, 2021, doi:10.3390/cells10061487_

Round 1

Reviewer 1 Report

The authors edited the manuscript to remove most of the grammatical errors and clarify some of the sentences, but they did not improve the scientific content of the manuscript.

The review lacks of a critical discussion of the published data that could make the review useful for the development of this field. The strength of the data reported in the review (Are the observations made in in vitro, ex vivo, in vivo experiments? Are they validated in different animal models? Are they confirmed in humans? Are the concentrations of the prostaglandins used in the studies achievable in vivo? Are the prostaglandin modulators used at clinical relevant concentrations?) was not discussed.

Moreover, the review missed to discuss pivotal papers in this field published in high impact journals like: J Clin Invest  2019, 129:3594-3609; Nature, 2019, 569:73-78. Nature, 2020, 580:524-529, just to name a few.

Author Response

We appreciate reviewer 1´s comment regarding the effort we have put in reviewing the grammatical errors and clarify some of the sentences. We have tried to further improve this last version, re-editing some of the contents and including new data. We have reedited the conclusions of this review and made a critical discussion about the type of experiments included in this work and other aspects We have put the best effort to check each reference to fix any mistake, and added new references including those the reviewer had suggested.

Reviewer 2 Report

The revision has improved.

Author Response

We really appreciate reviewer 2´s comment regarding that the manuscript has now improved. We have tried to further improve this last version, re-editing some of the contents and including new data and critical conclusions.

Round 2

Reviewer 1 Report

The authors addressed most of the reviewer`s concerns.

This manuscript is a resubmission of an earlier submission. The following is a list of the peer review reports and author responses from that submission.

Round 1

Reviewer 1 Report

The authors put a lot of effort in reviewing the literature related to the role of prostaglandin in different type of cancers, but unfortunately they were not able to write a comprehensive and scientifically sound review on this topic.

The review is not clearly written, it includes some sentences written in Spanish, it presents many grammatical errors and scientific terms improperly used.

There aren’t clear and common criteria for the selection of the papers reported in the review.

Important seminal and recent papers were not cited. Some references were not correct.

There isn’t a critical discussion of the data reported in the review, probably reflecting the lack of expertise of the authors on the topic covered by the review.

Reviewer 2 Report

In this review, the author reviewed functions of prostaglandins (PG) in different types of cancer. The paper is supported by huge body of literature and includes a lot of interesting data. They list the role of PG in different biological systems on tables and mentions action mechanism and development status, which might be interesting and helpful to readers. This review is comprehensive and well-organized. There are some minor adjustments required to be addressed. The specific comments are listed below:

  1. The role of prostaglandins is complex in different cancers, some are beneficial, and some are tumor-promoting. The title only shows the benefits may be not suitable.
  2. The use of “cancer” or “cancers” should be consisted.
  3. It is better to summary the results to figures, make it easy to catch the points.
  4. I am confused why show the same items several times in Table 2.
  5. There are some spelling and grammatical errors, the author should check it carefully.